

# Demographic, ecological, and physiological responses of ringed seals to an abrupt decline in sea ice availability

Steven H. Ferguson[1,2,3], Brent G. Young[1,2], David J. Yurkowski[1,2], Randi Anderson[2], Cornelia Willing[2,3] and Ole Nielsen[1]

[1] Fisheries and Oceans Canada, Winnipeg, MB, Canada
[2] Department of Biological Sciences, University of Manitoba, Winnipeg, MB, Canada
[3] Centre for Earth Observation Science, University of Manitoba, Winnipeg, MB, Canada

## ABSTRACT

To assess whether demographic declines of Arctic species at the southern limit of their range will be gradual or punctuated, we compared large-scale environmental patterns including sea ice dynamics to ringed seal (*Pusa hispida*) reproduction, body condition, recruitment, and stress in Hudson Bay from 2003 to 2013. Aerial surveys suggested a gradual decline in seal density from 1995 to 2013, with the lowest density occurring in 2013. Body condition decreased and stress (cortisol) increased over time in relation to longer open water periods. The 2010 open water period in Hudson Bay coincided with extremes in large-scale atmospheric patterns (North Atlantic Oscillation, Arctic Oscillation, El Nino-Southern Oscillation) resulting in the earliest spring breakup and the latest ice formation on record. The warming event was coincident with high stress level, low ovulation rate, low pregnancy rate, few pups in the Inuit harvest, and observations of sick seals. Results provide evidence of changes in the condition of Arctic marine mammals in relation to climate mediated sea ice dynamics. We conclude that although negative demographic responses of Hudson Bay seals are occurring gradually with diminishing sea ice, a recent episodic environmental event played a significant role in a punctuated population decline.

Corresponding author
Steven H. Ferguson,
steve.ferguson@dfo-mpo.gc.ca

## BACKGROUND

Organisms evolve specific adaptations to their habitats through natural selection (*Mayr, 1963*) and when their habitats change gradually, organisms can adjust phenotypically within an evolved range of flexibility (*Levins, 1962*). However, this evolved adaptation has limitations and in extreme situations, organisms may not be able to adapt to particular habitats and environmental conditions above an evolved threshold (*Southwood, 1977*). Under these circumstances, populations suffer mortality of individuals, declines in reproduction, and/or immigrate to new habitats that may allow increased demographic success (*MacArthur & Wilson, 1967*). The result is a shift in species distribution (*Guisan & Thuiller, 2005*) and understanding this process by identifying

thresholds to adaptability and the mechanism of population decline are both critical to species conservation.

Predicting how climate warming will result in retraction of an Arctic species range northward requires knowledge of demographic changes and their ecological plasticity in response to environmental change. Few studies have linked marine mammal demographic responses to climate change (*Poloczanska et al., 2007*) with the notable exception of polar bears (*Ursus maritimus*) (*Regehr et al., 2007*; *Hunter et al., 2010*; *Lunn et al., 2016*), where the majority of research relates to loss of space and time opportunities for feeding on a lipid-rich diet (*Thiemann, Iverson & Stirling, 2008*; *Rode et al., 2016*). Ringed seals (*Pusa hispida*) have a circumpolar distribution and show high variability in the relative importance of predation from polar bears (*Thiemann, Iverson & Stirling, 2008*) and to varying food habits (*Yurkowski et al., 2016b*). However, key habitat attributes are linked to survival and successful reproduction. In particular, ringed seals require sea ice during the critical spring period when reproduction and molting occurs (*Smith & Stirling, 1975*) and a seasonal pulse in food availability in the summer ice-free season (*Young & Ferguson, 2013*). Evolved life history characteristics that match these high-latitude environmental features include relative small body size for a pinniped and a life history characterized by early age of maturation, annual birthing, short lactation duration, widely varying but high pup mortality, relatively low adult mortality, and greater fitness investment in long life (*Ferguson & Higdon, 2006*).

High latitude species are characterized by a strong seasonal cycle of feast and fast with both periods critical to reproduction and survival (*Boyce, 1979*). Ringed seals are adapted to cycle annually from intensive foraging during the open water season to accumulate fat reserves to sustain them over winter and during the birthing, nursing, and mating periods when adults are restricted to small home ranges with depleted food resources (*Luque, Ferguson & Breed, 2014*; *Young & Ferguson, 2014*). In spring, pups are independent and adults undergo molting with little feeding opportunities and increased risk of predation (*Stirling & Archibald, 1977*). During periods of deteriorating environmental conditions, the phenology of ringed seals can be interrupted leading to inadequate energy reserves prior to the next year's reproduction (*Harwood et al., 2012*). Ringed seal populations can also be negatively affected by infrequent, annual, extreme climatic conditions that exert pressure on their demographics (*Stirling & Smith, 2004*).

Endemic Arctic species are challenged by the rapid pace of sea ice declines and resulting changes in ecological dynamics of the marine ecosystem (*Post et al., 2013*). Hudson Bay represents one of the most southerly distributions of ringed seals and therefore, as an ice-obligate marine mammal, the prediction is for a retraction northward in range (*Kovacs & Lydersen, 2008*). The Hudson Bay ecosystem is at the southern edge of maximum sea ice extent and goes through a seasonal cycle of complete ice formation and loss (*Saucier et al., 2004*). The initial characteristics of population and demographic changes may already be occurring with a decrease in ringed seal density observed in western Hudson Bay between the two recent aerial surveys in spring 2010 and 2013 ($0.78$–$0.20$/km$^2$) (*Young, Ferguson & Lunn, 2015*).

Here, we compare indices in the productivity and health of the Hudson Bay ringed seal population with environmental covariates over time. Our data sets were annual trends in sea ice breakup and formation, major climatic indices, and biological data from seal collections, 2003–2013: (1) body condition (% fat) from seals harvested by Inuit; (2) reproduction from examination of reproductive tracts; (3) recruitment from hunter harvest statistics; and (4) stress from blubber cortisol levels. We hypothesize that gradual deteriorating change in sea ice characteristics will correlate with a gradual decrease in ringed seal body condition, ovulation rate and pup recruitment (*Stirling, 2005*), whereas an abrupt decline in sea ice availability in 2010 will result in dramatic negative demographic, ecological, and physiological responses by ringed seals.

## METHODS

Sea ice breakup and freeze-up dates were determined from weekly data obtained from the Canadian Ice Service using Icegraph 2.0 (http://iceweb1.cis.ec.gc.ca/IceGraph/), for eastern Hudson Bay, 1979–2014. The majority of biological data for ringed seals was derived from the Sanikiluaq (southeast Hudson Bay) seal collection whereas Arviat (southwest Hudson Bay) only provided the time-series of cortisol measures. Therefore, we only present annual changes in sea ice coverage (Fig. 1B) for this eastern region (Sanikiluaq) although we found strong correlations with other Hudson Bay regions identified by the Canadian Ice Service. We were unable to assess the effect of east-west differences (*Young & Ferguson, 2014*) and how they may influence our results because our datasets were not balanced (i.e., biological measures from Sanikiluaq were not available from Arviat). Ice breakup date was defined as the date on which the sea ice concentration decreased and remained below 50% (*Stirling, Lunn & Iacozza, 1999*). Conversely, freeze-up date was defined as the date on which sea ice concentration increased and remained above 50%. Open-water duration was calculated by subtracting the breakup and freeze-up dates. Major climatic indices were obtained from the Climate Prediction Center (http://www.cpc.ncep.noaa.gov/), including the Arctic Oscillation (AO), the North Atlantic Oscillation (NAO), and El Nino-Southern Oscillation (ENSO) for the December to February monthly mean estimates from 1971 to 2014. We included ENSO due to its significant climatic influence in North America and due to its effect on ecological relationships in several ecosystems across the globe (*Wang et al., 2010*; *Nye et al., 2014*; *Rustic et al., 2015*). NAO and AO were included since previous research found that they were related to ringed seal recruitment and timing of spring ice breakup (*Ferguson, Stirling & McLoughlin, 2005*). The longer time frame available for environmental data provided a background to the 2003–2013 period with available ringed seal biological data.

Morphological measurements and tissue samples were collected from 1,425 Hudson Bay ringed seals harvested during the Inuit subsistence hunt from Sanikiluaq ($n = 917$), NU, Canada (56°32′34″N, 79°13′30″W) and Arviat ($n = 508$), NU (61°6′29″N, 94°3′25″W) from 2003 to 2013. Permits to collect samples as part of the Inuit subsistence hunts were acquired from Fisheries and Oceans Canada. All biological data with the exception of cortisol was derived from seals collected by Sanikiluaq hunters in the eastern region of

Hudson Bay. Samples were collected in autumn (October to December) in Arviat. In Sanikiluaq, some samples were collected throughout the year but we used only autumn collections for age/sex composition as the late open-water season provides a representative sample of the population (see *Holst, Stirling & Calvert, 1999*; *Ferguson, Stirling & McLoughlin, 2005*). Permits to collect samples as part of the Inuit subsistence hunts were acquired from Fisheries and Oceans Canada. Canine teeth were extracted from the lower jaw for age determination using annual growth layer groups in the cementum (*Chambellant & Ferguson, 2009*). Pup survival was defined as the percentage of pups (i.e., <1 year) in the autumn subsistence hunt and is considered a good measure of 0–6 month survival (*Chambellant et al., 2012*). Total body weight and sculp weight (weight of blubber layer, skin, and fur) were recorded by the hunters at the time of sample collection. Body condition was calculated as percent blubber (sculp weight × 100/total bodyweight). Reproductive tracts were stored frozen before being examined. After gross examination of reproductive tracts, ovaries were excised, formalin-fixed, and sectioned at 2-mm intervals, and examined macroscopically for the presence of a corpus luteum (ovulation in the year of collection) and corpora albicantia (previous pregnancies) (*Laws, 1956*). Estimation of ovulation was only calculated if sample size for a particular year exceeded five mature adult females which excluded 2003–2006. An extraction method for ringed seal blubber samples was used in conjunction with radioimmunoassay to measure cortisol levels representing stress (*Trana et al., 2014*). Cortisol measures from Arviat seals were not available in 2013.

Four separate general linear models were used to investigate relationships between environmental (i.e., duration of the open water period, ENSO, NAO, and AO indices) and biological variables (i.e., percentage of ovulating females, percentage of pups in the harvest, body condition, and cortisol levels) over time using R v 3.2.3 (*R Core Team, 2015*). Continuous predictor variables were screened for collinearity and removed when a Pearson's correlation coefficient was ≥0.6 and a variance inflation factor was >3.0. NAO and AO were highly correlated (0.8). We retained NAO for all analyses due to its stronger association with sea ice (*Nakamura et al., 2015*). Prior to analysis, percentage of ovulating females, percentage of pups in harvest, and body condition were normally distributed upon visual examination of histograms and quantile–quantile plots. Cortisol levels were log-transformed before analysis to improve normality.

## RESULTS

Results support a gradual pattern of earlier spring ice breakup and later autumn freeze-up in Hudson Bay; where from 2003 to 2013, sea ice breakup has varied more widely than freeze-up. No relationship occurred with any climate variability index over 1979–2014, but the NAO and AO have been more positive from 1999 to 2015 (Fig. 1). The longest ice-free season on record for eastern Hudson Bay occurred in 2010, with the earliest spring breakup (May) and latest freeze-up (January 2011) and an anomalous negative NAO and AO, and a high ENSO index (Fig. 1).

Body condition significantly decreased over time ($t = -8.2$, $p < 0.001$), from 55.4% blubber mass in 2004 to 40.3% in 2012 before increasing to 48.1% in 2013 (Table 1; Fig. 2).

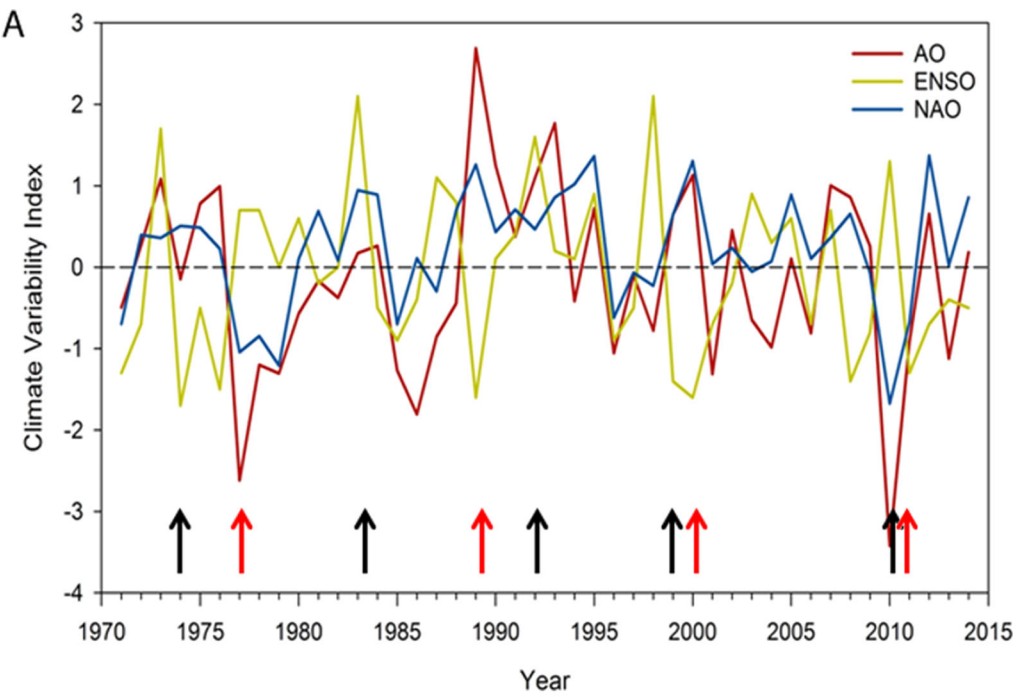

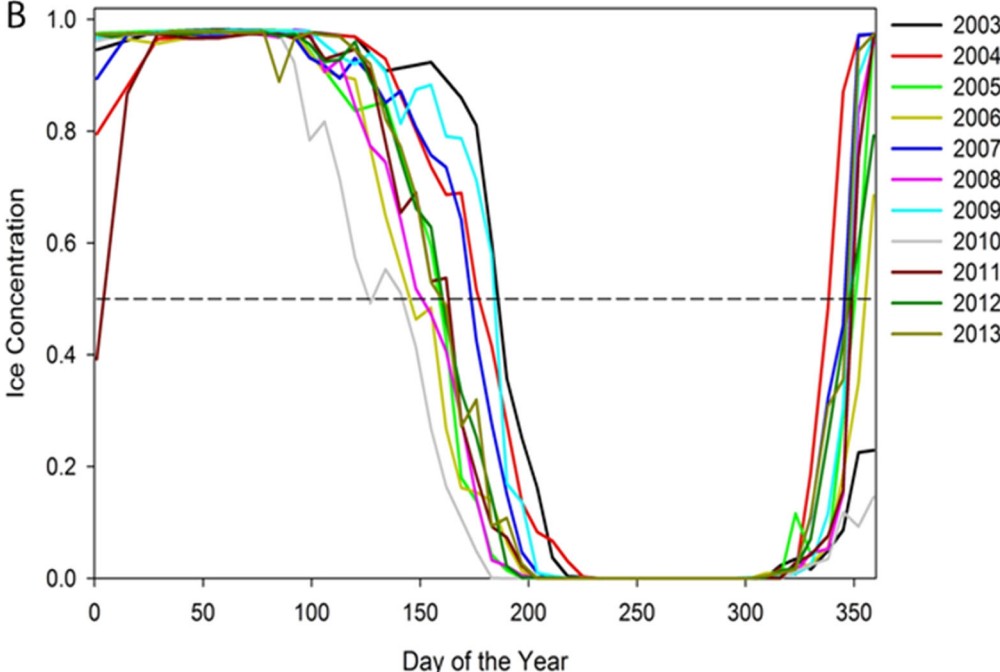

**Figure 1** **(A) Annual winter (December to February) North Atlantic Oscillation index (NAO), Arctic Oscillation (AO), and El Nino-Southern Oscillation (ENSO), 1971–2014.** Note red arrows indicate possible regime shifts (1977, 1989, 1989/99, 2010) and black arrows possible years with poor ringed seal condition: 1973/74, 1983, 1992, 1998, 2010 (*Smith & Stirling, 1978*; *Hare & Mantua, 2000*; *Smith & Harwood, 2001*; *Benson & Trites, 2002*; *Ferguson, Stirling & McLoughlin, 2005*; *Litzow, 2006*). (B) Sea ice patterns over the day of the year showing inter-annual variation in timing of spring breakup, duration of open water season, and time of freeze-up, 2003–2013. Note that autumn 2010 freeze-up did not occur until January 2011.

**Table 1 Biological data from harvested ringed seals collected in Hudson Bay, Canada.**

| Year | Body condition (% fat) | Ovulation (%) | Young of year (%) | RIA cortisol (ng/g) |
|------|------------------------|---------------|-------------------|---------------------|
| 2003 |                        |               | 39.1 (115)        | 0.07 ± 0.01 (72)    |
| 2004 | 55.4 ± 1.0 (45)        |               | 30.4 (56)         | 0.10 ± 0.02 (32)    |
| 2005 | 52.8 ± 0.9 (71)        |               | 40.9 (88)         | 0.10 ± 0.01 (120)   |
| 2006 | 49.7 ± 0.7 (80)        |               | 42.7 (82)         | 0.14 ± 0.05 (40)    |
| 2007 | 46.9 ± 0.6 (123)       | 85.7 (7)      | 52.4 (126)        | 0.32 ± 0.10 (27)    |
| 2008 | 47.5 ± 0.8 (102)       | 100.0 (5)     | 48.6 (105)        | 0.24 ± 0.04 (56)    |
| 2009 | 45.2 ± 1.2 (41)        | 88.9 (9)      | 38.1 (42)         | 0.28 ± 0.04 (51)    |
| 2010 | 43.9 ± 0.9 (90)        | 66.7 (18)     | 28.1 (96)         | 0.86 ± 0.27 (46)    |
| 2011 | 46.4 ± 0.8 (97)        | 56.3 (16)     | 20.6 (97)         | 0.51 ± 0.12 (30)    |
| 2012 | 40.3 ± 1.2 (65)        | 83.3 (12)     | 10.8 (65)         | 0.43 ± 0.07 (34)    |
| 2013 | 48.1 ± 1.4 (42)        | 100.0 (6)     | 20.0 (45)         |                     |

Note:
Mean ± standard error (sample size).

In addition, body condition significantly decreased with increasing open water period ($t = -2.0$, $p < 0.05$), ENSO index ($t = -2.3$, $p = 0.02$), and NAO index ($t = -2.0$, $p < 0.05$; Table 2; Fig. 3). Ovulation rate varied considerably among years from 100% in 2008 to 56% in 2011, albeit with no relationship with year, open water duration, or climatic indices. Percentage of pups in the harvest, as an estimate of pup survival, exhibited a marginal decline from 2003 to 2013 ($t = -2.09$, $p = 0.08$) from about 40% of the harvest to about 20% (Table 2; Fig. 2). Stress, as measured by cortisol concentration (ng/g), significantly increased over time ($t = 8.0$, $p < 0.001$) from about 0.1 to 0.6 ng/g over the 2003–2012 period (Table 2; Fig. 2). A significant decrease in cortisol level occurred with NAO index ($t = -2.6$, $p = 0.01$; Fig. 3). In 2010, cortisol levels in ringed seals had the highest amount of variability (standard deviation = 1.84) compared to other years (Fig. 2). High stress levels occurred in 2010 and low ovulation rates occurred in 2011 which supports the pattern of a decrease in ovulation rate after the high stress levels.

## DISCUSSION

We predicted demographic change occurring at the southern limit of the ringed seal distribution with both gradual changes in environmental variables and episodic events associated with extreme lows in sea ice concentration. Our results suggest both patterns have occurred in southern Hudson Bay over the past decade. Previous research has indicated that Hudson Bay ringed seals (*Chambellant et al., 2012*) and polar bears (*Regehr et al., 2007*; *Lunn et al., 2016*; *Obbard et al., 2016*) have shown gradual reductions in body condition and survival over the past decades which are concurrent with negative consequences of continued environmental change (*Holst, Stirling & Calvert, 1999*; *Ferguson, Stirling & McLoughlin, 2005*). We provide additional evidence for a continuation of these progressive patterns for ringed seals with decreasing body condition and increasing stress over 2003–2013. However, no research results have suggested short-temporal pulses in condition and abundance (i.e., *Young, Ferguson & Lunn, 2015*) for

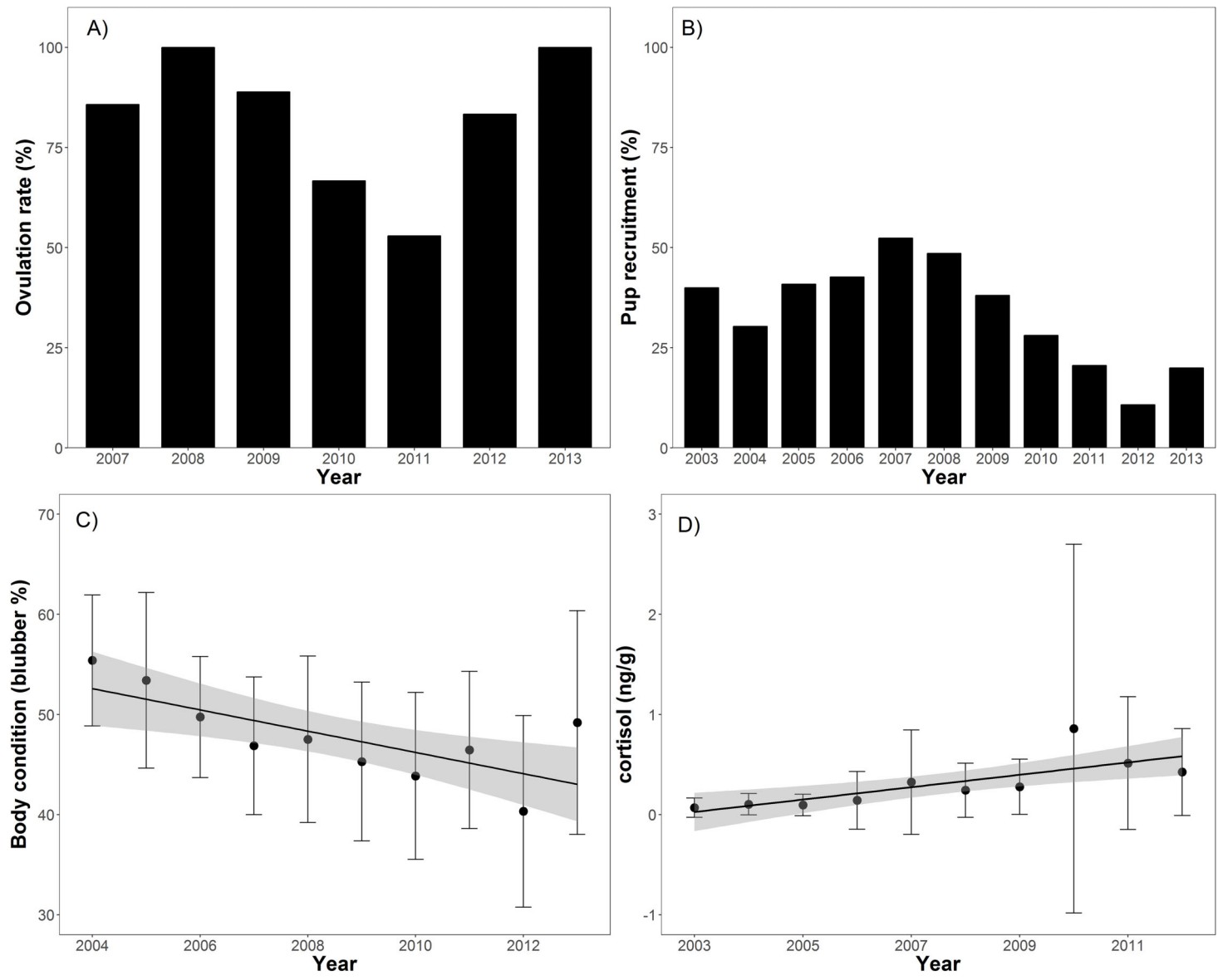

**Figure 2 Barplots (A and B) of annual ovulation rates (%) from adult female ringed seals and annual percentage of pups in the harvest (Table 1).** Linear regressions between seal body condition and harvest year (C; slope = −0.01, *t* = −8.2, *p* < 0.001), and cortisol level and harvest year (D; slope = 0.02, *t* = 8.0, *p* < 0.001).

either seals or polar bears in the Hudson Bay ecosystem, although a regime shift likely occurred in late 1990s (*Gaston, Smith & Provencher, 2012*). Here, we document for the first time, a relationship with ringed seal demographics and the 2010 climatic event that resulted in a punctuated decrease in ovulation, reduced body condition, and increased cortisol levels. Reduced seal pups in the following autumn harvest would likely follow with a lag effect (*Ferguson, Stirling & McLoughlin, 2005*; *Iacozza & Ferguson, 2014*). Ringed seals display a remarkable ability to adjust their body condition and reproduction with different environmental conditions as exemplified by the return to high ovulation levels and body condition (% fat) in the years following the 2010 extreme event. However, age

**Table 2 Relationships between Hudson Bay ringed seal biological parameters and environmental correlates assessed using general linear models, 2003–2013.**

| Covariates | Ovulation rate (%) | Pup recruitment (%) | Seal condition (blubber %) | Cortisol (ng/g) |
|---|---|---|---|---|
| Intercept | −48.00 ± 69.51 | 41.96 ± 39.1 | 25.20 ± 2.58** | −34.90 ± 43.48** |
| Year | 0.02 ± 0.03 | −0.021 ± 0.020 | −0.012 ± 0.001** | 0.002 ± 0.0002** |
| Ice-free period (days) | 0.0003 ± 0.006 | −0.00036 ± 0.0033 | −0.0003 ± 0.0002[a] | 0.00003 ± 0.0005 |
| El-Niño Southern Oscillation | 0.004 ± 0.01 | 0.0175 ± 0.072 | −0.008 ± 0.004* | 0.001 ± 0.006[b] |
| North Atlantic Oscillation | 0.011 ± 0.012 | 0.00719 ± 0.072 | −0.009 ± 0.004* | −0.02 ± 0.007* |

Notes:

$R^2$ was 0.40 for ovulation rate model, 0.19 for pup recruitment model, 0.12 for body condition model, and 0.21 for cortisol level model.

*$p < 0.05$; **$p < 0.001$.

[a] 0.08.

[b] 0.055.

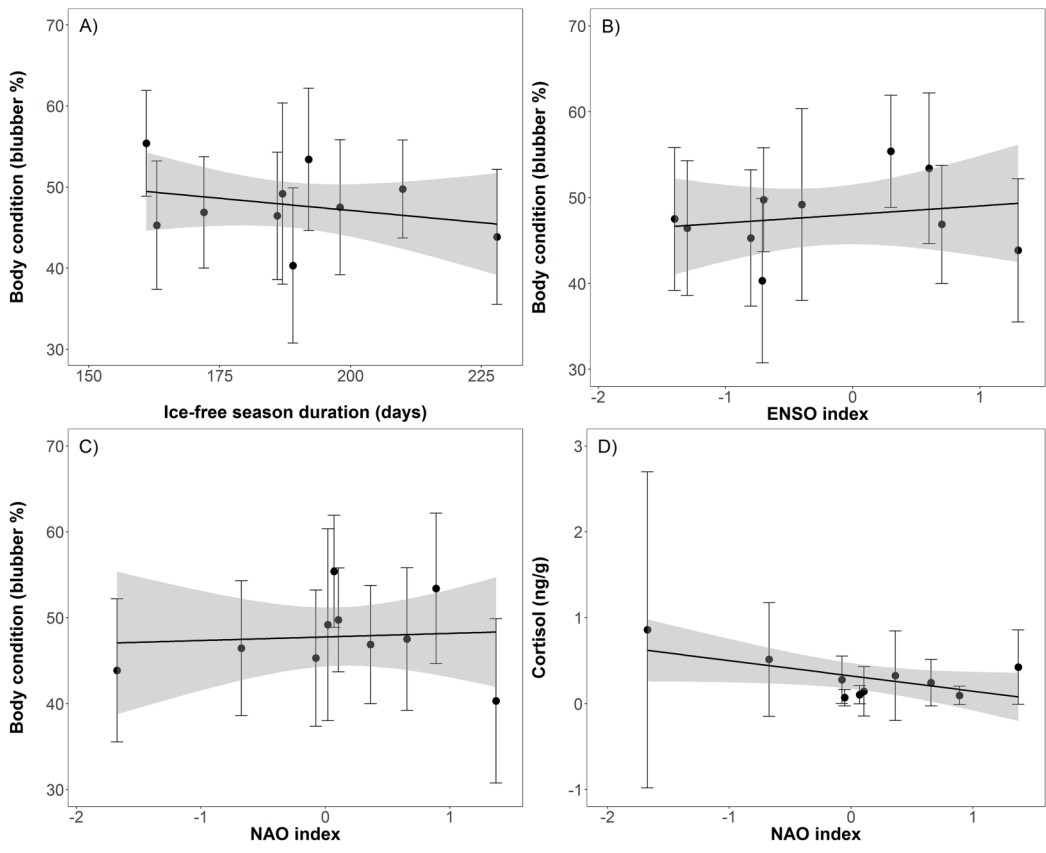

**Figure 3 Linear regressions between ringed seal body condition and ice-free duration (A; slope = −0.0004, $t = -2.0$, $p < 0.05$), body condition and El-Nino Southern Oscillation (ENSO) index (B; slope = −0.009, $t = -2.32$, $p = 0.02$), body condition and North Atlantic Oscillation (NAO) index (C; slope = −0.009, $t = 2.0$, $p < 0.05$), and cortisol and NAO index (D; slope = −0.02, $t = 2.6$, $p = 0.01$).**
structure would likely maintain a record of a cohort effect with a reduced number of seals moving through the population over time.

Gradual reduction in body condition could be associated with the recent changes in Hudson Bay prey resource abundance and availability. The prevalence of capelin (*Mallotus villosus*) and sand lance (*Ammodytes* spp.) and decrease in Arctic cod (*Boreogadus saida*) abundance in Hudson Bay since 2000 has caused dietary shifts from endemic Arctic cod to sub-Arctic capelin and sand lance in Arctic marine megafauna including sea birds (*Gaston, Woo & Hipfner, 2003*), beluga whales (*Delphinapterus leucus*; *Kelley et al., 2010*), and ringed seals (*Chambellant et al., 2012*). In addition, the isotopic niche size of Hudson Bay ringed seals is significantly larger than individuals from higher latitudes which principally consume Arctic cod, indicating a more diverse and omnivorous diet (*Young & Ferguson, 2013*; *Yurkowski et al., 2016a*, *2016b*). Among ringed seal prey items, Arctic cod represent the highest energy content compared to other fish and invertebrate species (*Weslawski et al., 1994*; *Hedeholm, Grønkjær & Rysgaard, 2011*; *Harwood et al., 2015*). Thus, a recent change in Hudson Bay ringed seal diet due to shifts in forage fish availability and abundance may have negatively impacted ringed seal body condition.

Assessing the causes of an episodic event is more difficult to establish. The extremely low extent and duration of the 2010 ice-covered period in Hudson Bay may have adversely affected the abundance, availability and distribution of prey resources but it is unlikely to have triggered a punctuated decrease in their physiological and energetic demands. We summarized anecdotal evidence for an episodic event affecting the abundance and body condition of ringed seals in Hudson Bay related in 2010–2011 (see Supplemental Material). Anecdotal observations in 2010 are suggestive of a hitherto never before seen event causing impaired biological responses in ringed seal behavior including unusual approachability, lethargy, and increased tendency for hauling out on land, possibly due to associated respiratory problems that were first seen during that autumn season. Polar bears are thought to have benefited from this behavior since affected seals were easily captured but no estimate of predation over and above normal could be calculated. Evidence for a biological response to an episodic environmental event comes from the low ringed seal density observed between spring 2010 and 2013 surveys and the unusual environmental patterns that suggest a possible shift in seal condition after 2010.

Evidence for a dramatic decline in ringed seal abundance associated with the 2010 climatic event in Hudson Bay is both anecdotal (Table S1) and circumstantial (aerial survey abundance estimates; *Young, Ferguson & Lunn, 2015*). The mechanism of such a decline is not well understood but we postulate that it may be linked to the inability of the seals to properly molt in spring due to a lack of a resting platform with the early loss of sea ice which sets up a physiological predisposition for disease. In addition, hyperthermia in autumn when seals are at their maximum blubber fatness (*Young & Ferguson, 2013*) may be a potential mechanism for the observations of lethargy and use of tidal flats resulting in greater polar bear predation (Table S2). The evidence for a decline in Hudson Bay ringed seal body condition from 2003 to 2013 has statistical support and continues a pattern previously reported (*Stirling, 2005*). Periods of declines in ringed seal body condition have been documented in the western Canadian Arctic (*Harwood et al., 2012*) and

Svalbard (*Hamilton et al., 2015*) as well as periods of improving body condition in western Hudson Bay (*Chambellant et al., 2012*). In all cases, top down predation is not considered the agent of change but rather bottom up changes in food supply.

Longer periods of open-water have been linked to access to more food for ringed seals allowing for a longer period of fattening (*Young & Ferguson, 2013*). Possible explanations for this novel pattern of decreased ringed seal condition with a warming ocean include: (1) a shift in the types of forage fish available that result in lower lipid intake—a requirement for ringed seals with their large blubber biomass (*Gaden et al., 2009*; *McKinney et al., 2013*; *Yurkowski et al., 2016a*, *2016b*); (2) greater competition from temperate species making forays into the subarctic (*Finley, Bradstreet & Miller, 1990*; *Berg et al., 2010*); (3) greater predation effect from new predators moving into areas from which they were previously excluded by sea ice forcing ringed seals to compromise foraging activities in favor of predator avoidance (*Laidre et al., 2008*; *Higdon & Ferguson, 2009*); and (4) new or increased disease arising from physiological stress associated with warmer temperatures (*Pounds et al., 2006*; *Burek, Gulland & O'Hara, 2008*). For Hudson Bay we found ringed seal condition problems linked to large-scale climatic patterns that likely cycle over multiple years and possibly explain the periods of good (*Chambellant et al., 2012*) and bad (this study) in Hudson Bay. However, the mechanistic link between early spring breakup and late ice formation and poor seal condition is not well understood possibly because it has been rarely observed (*Ferguson, Stirling & McLoughlin, 2005*).

An Unusual Mortality Event was declared in 2011 by the US government due to a "new" ulcerative–dermatitis-disease-syndrome of unknown etiology observed in Alaskan ice seals and Pacific walrus (*Atwood et al., 2015*) that resulted in significant pathology of the lung, liver, immune system, and skin of the seals (*Barbosa et al., 2015*; *Bowen et al., 2015*). As observed in Hudson Bay, the affected ice seals displayed uncommon behaviors such as unusual approachability, lethargy, and increased tendency for hauling out on land, as well as respiratory problems. There was some mortality associated with the disease syndrome; however reliable baseline abundance estimates were not available to assess its significance.

Potential repercussions of a gradual sea ice decline and punctuated decreases in some years include a continual reduction in ringed seal body condition and greater stress leading to implications on their demographics. The years marked by extremes in climatic indices (Fig. 1) are associated at higher latitudes with excessive sea ice extremes; whereas our results at the southern range of ringed seals indicate a lack of sea ice may have attributed to decreased body condition, increased stress, and low ovulation rates and pup recruitment. Spring 2010 recorded an unusually early ice breakup that may have predisposed seals to a delayed molt. In the fall of 2010, numerous (100s) moribund seals were found in distress along the shore of western and eastern Hudson Bay suggesting that both regions were affected.

Numerous examples of episodic events causing major ecological shifts include regime shifts (*Hughes et al., 2013*), continental growth (*Santosh, 2013*), drought (*Ireland et al., 2012*), disease (*Pickles et al., 2013*), and range shifts due to climate (*Baker, Glynn & Riegl, 2008*; *Seppä et al., 2009*; *Chen et al., 2011*). For ringed seals, the literature suggests

periods of ringed seal crashes in abundance associated with poor reproduction during significant heavy ice years. Variation in ringed seal density associated with ENSO events include 1973 (*Smith & Stirling, 1978*), 1992 (*Ferguson, Stirling & McLoughlin, 2005*), 1998 (*Smith & Harwood, 2001*), and in 2010 (Fig. 1). Evidence of high latitude regime shifts include 1977 and 1989 (*Hare & Mantua, 2000*), 1998–1999 (*Litzow, 2006*; *Benson & Trites, 2002*). Also, the Greenland Blocking Index for 2010 was the highest year in the annual, spring, winter, and December series, 1851–2015 (*Hanna et al., 2016*). Synchronous fluctuations of seabird species across the entire Arctic and sub-Arctic regions were associated with changes in sea surface temperatures that were linked to two climate shifts, in 1977 and again in 1989 (*Irons et al., 2008*), and 1998 (*Flint, 2013*), including Hudson Bay in 1998 (*Gaston, Woo & Hipfner, 2003*). Major atmospheric patterns suggest that we can expect episodic events occurring once every 10–15 years and that they are largely unpredictable in timing but have major consequences on ecosystem structure and function (*Ottersen, Stenseth & Hurrell, 2004*).

## CONCLUSION

Considerable uncertainties exist with deciphering past patterns to determine possible cause and effect relationships among environmental variation, body condition, and their demographic responses. However, mounting evidence indicates endemic Arctic species, such as ringed seals, are under immense pressure from climate change and complex spatio-temporal shifts in ecology have subsequently resulted in decreased abundance as a harbinger of range shift. Managers need to be wary of climate change culminating in both a gradual decline in condition and unpredictable episodic events that when combined can have major abundance and distribution consequences.

## ACKNOWLEDGEMENTS

We thank the Inuit hunters and the Hunters and Trappers Association of Arviat and Sanikiluaq, NU, Canada, for conducting community-based seal collections. Reviews by J. Higdon, R. Hodgson, N. Pilfold, and two anonymous reviewers improved the manuscript.

### Funding

Funding was provided by the Natural Sciences and Engineering Research Council (#1025) of Canada, Federal Program Office of International Polar Year (MD-112), Nunavut Wildlife Management Board (#3-09-04), ArcticNet (#317588), and Fisheries and Oceans Canada (NIF-05). The funders had no role in study design, data collection and analysis, decision to publish, or preparation of the manuscript.

### Grant Disclosures

The following grant information was disclosed by the authors:
Natural Sciences and Engineering Research Council: #1025.
Federal Program Office of International Polar Year: MD-112.

Nunavut Wildlife Management Board: #3-09-04.

ArcticNet: #317588.

Fisheries and Oceans Canada: NIF-05.

## Competing Interests

The authors declare that they have no competing interests.

## Author Contributions

- Steven H. Ferguson conceived and designed the experiments, analyzed the data, contributed reagents/materials/analysis tools, wrote the paper, prepared figures and/or tables, reviewed drafts of the paper.
- Brent G. Young performed the experiments, analyzed the data, wrote the paper, prepared figures and/or tables, reviewed drafts of the paper.
- David J. Yurkowski analyzed the data, wrote the paper, prepared figures and/or tables, reviewed drafts of the paper.
- Randi Anderson performed the experiments, reviewed drafts of the paper.
- Cornelia Willing performed the experiments, reviewed drafts of the paper.
- Ole Nielsen performed the experiments, contributed reagents/materials/analysis tools, reviewed drafts of the paper.

## Animal Ethics

The following information was supplied relating to ethical approvals (i.e., approving body and any reference numbers):

Permits to collect samples as part of the Inuit subsistence hunts were acquired from Fisheries and Oceans Canada.

## Data Deposition

The raw data has been supplied as Supplemental Dataset Files.

## Supplemental Information

Supplemental information for this article can be found online at http://dx.doi.org/10.7717/peerj.2957#supplemental-information.

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
