# Peer review of "Demographic, ecological, and physiological responses of ringed seals to an abrupt decline in sea ice availability"

_PeerJ, doi:10.7717/peerj.2957_

## Round 0.1 · original submission · Major Revisions

· Academic Editor

Major Revisions

The three reviewers have each provided detailed suggestions for the improvement of the manuscript. Please address each of these suggestions in any revised manuscript you submit along with the rebuttal letter.

Reviewer 1 ·

Basic reporting

see below

Experimental design

see below

Validity of the findings

see below

Additional comments

1. Line 106 – it should be made clear here that the 926 ringed seals were adult females. It isn’t until one adds up the numbers in the figure 2 legend that it is absolutely clear that you only sampled females for body condition and cortisol.

2. There can be an issue of bias with sample collection from subsistence hunting which you have not addressed anywhere in the manuscript. Generally animals in poor condition will not be targeted and, thus, your sample is not random – hunters take the best of what is available to them. This is not a criticism of the sampling – it is an unavoidable reality for this kind of work – but it does need to be noted somewhere in the manuscript since it influences the samples you are analysing – you will always be getting animals from the high end of the body condition scale, regardless of the available overall sample of animals which means the changes you observe in your samples may be less dramatic than the reality.

3. As presented the sample sizes for each of your analyses are unclear. Either data is missing from panels C and D in Figure 2 or there are errors in these plots. For Panel 2C – was body condition data not collected in 2003 or is there data missing from the plot? If it was not collected please update the description in the methods to indicate that there is no data for this parameter for that year and edit Fig 2C so that the data by year in 2C aligns with the years in 2A above (2C should start at 2003). Likewise for Panel 2D – was cortisol not measured in 2013 or is there data missing from the plot? If it was not collected please update the methods section to make that fact clear and edit the panel so that the years in 2D align with 2B. Also, please verify that the numbers given in the figure legend apply to panels A,C and D - if all females were not sampled for all components (ovulation rate, body condition and cortisol) in all years then the numbers for each panel need to be outlined somewhere (in the panels or in a table) so that the sample sizes for each of the components are clear.

4. Please indicate in the methods why is there no ovulation rate data for 2004-2006.

5. Lines 111 – 113. What was the age data used for? If it was not entered in any of the models then there is no reason to mention its collection. If it was used in the models then it is missing from the description.

6. How was body condition (blubber %) determined? I can’t find a description of how the value was obtained anywhere in the methods. Is this value a % fat content of a sample of blubber, is it an estimate of total fat content of the carcass or is it the % of the total carcass mass which was blubber only? This is an important measurement for your overall analysis and a description of how it was obtained is absolutely necessary here.

7. Line 128 – based on the results presented I assume it was AO that was removed from analysis. Sentence should read ‘…..thus, AO was removed from all analyses.’ As written it suggests the opposite.

8. Line 141 – Why do you use 2011 as the lower reference (~45%) here when 2012 is even lower at ~40%? Also, body condition increases rather substantially in 2013 relative to 2012 but you make no reference to the increase – why?

9. Figure 3 – it would be helpful to have the points in the figure coded by year (different symbols &/or shading) – it would make them easier to relate to figure 2C and 2D.

10. Figure 3B – why are there only 9 data points? Assuming data for body composition was not collected in 2003 there should be 10 points. It appears that the body condition from 2012 (the lowest value) is missing. Is there an error in the plot or was that year removed from the analysis?
Lines 150-151. Please remove reference to ‘marginally significant’. You chose a cut-off of 0.05 for your definition of significant. The p value is above that and is, therefore, not significant.

11. Lines 153-155. According to Figure 2C, the lowest ovulation rates were in 2003 not 2011…..

12. Your data indicate a dramatic increase in body condition in 2013 (return to 2006 levels) which you do not address. If you are arguing a continued progressive decline in this parameter then you need to address the clear upswing at the end – why do you think it may have occurred and how does it fit with your argument? Similarly with ovulation rate, which also shows a dramatic increase in 2012 (the year of lowest body condition) with a continued increase in 2013 – how does this fit with your hypothesis?

13. Line 169-1172. Here you are suggesting that the 2010 conditions resulted in a punctuated decrease in ovulation, body condition and proportion of seal pups in the harvest. However, based on your figure 2, ovulation rate had been declining since 2008 and the proportion of pups had been declining since 2007. Body condition was on the low end of the scale in 2010 but it the change from 2008 to 2009 is no less dramatic than the change from 2009 to 2010 and body condition in 2012 is actually lower than 2010. I’m afraid I do not see how your data supports a punctuated change in relation to 2010 conditions. Your data supports an overall decline in pup recruitment (although variability is high) which is, in and of itself, an important finding but, as presented, the evidence for a punctuated decrease in the parameters listed in relation to a single year is very poor.

Reviewer 2 ·

Basic reporting

This manuscript reports on trends in ringed seal demography, body condition and physiology in Hudson Bay using samples collected from harvest monitoring programs in two Inuit communities in the Canadian Arctic. The authors present analysis on the relationships between sea ice conditions and environmental variables in relation to ovulation rates, young of the year in the open water harvest, body condition and cortisol stress values. The manuscript documents long-term declines in ringed seal body condition and increasing levels of cortisol stress which supports previous observations of changes in the condition of Arctic marine mammals in relation climate mediated sea ice dynamics. In regards to basic reporting it is worth noting that the authors of this paper fail to reference two key papers that relate directly to the results of their research (Harwood et al. 2012. Ringed seals and sea ice in Canadas western Arctic: harvest-based monitoring 1992-2011 and Stirling 2005. Reproductive rates of ringed seals and survival of pups in Northwestern Hudson Bay, Canada, 1991–2000). Both papers appear in the journal Arctic and measure several of the same variables used in this study (e.g. body condition and young-of-the-year in open water harvest (YOY)). In particular, Stirling 2005 who also collected seals in Arviat documents variable rates in YOY in the open water harvest that need to interpreted in association with the results presented herein. More specifically, despite large sample sizes Stirling 2005 documented low rates of young of the year in the open water harvest in years in which the ice-free period was likely much shorter than that reported by the authors in 2010.

In general I found the paper easy to follow and it read very well. However, I did find the inclusion of the anecdotal observational data from 2010 and the discussion surrounding the potential importance of a disease epidemic of some kind a little disjunct from the objectives identified in the background portion of the manuscript. This topic took up a significant portion of the discussion and seems to me to detract from the value of the data set which link demography, physical condition and stress. Yes, stress associated with environmental change could have led to compromised immune function and an increase in the susceptibility of seals to some unknown pathogen but there is no evidence provided to support this and therefore all discussion on the topic are purely circumstantial and speculative in nature. As a result I would suggest the authors significantly reduce this portion of the manuscript.

Experimental design

The authors provide analysis on original primary research that is not published elsewhere and do an adequate job of identifying their research objectives in the background of the document. There are some shortcoming in the methods section in regards to how certain variables were calculated (e.g. open water period and blubber percent) which the reader is left to assume. The choice of large scale climate variables (AO, NAO, & ENSO) in this analysis however does leave the reader wondering what the linkage is between these continental level weather patterns and the regional scale factors that influence ringed seal life history in the Hudson Bay ecosystem (e.g. what is the relationship between NAO and sea ice in Hudson Bay).

Validity of the findings

The authors analyze a variety of data sources in the manuscript using general linear models including three dependent variables that are presented as percentages. One of the statistical issues that results from using percentage data is that the data are bounded (i.e. a limited range variable) and therefore linear predictions beyond 0 and 100 percent are not valid. This is typically not a problem if most of the data fall between 30% and 70% but this is not the case for the ovulation rate data. There are several analytical approaches that can be taken to deal with this including data transformations and beta regression which the authors should investigate.
The following are more detailed comments relating to the assumptions and interpretation of the results of the study.
Line 93-94: Could the authors provide some indication of how representative sea ice trends in eastern Hudson Bay as cited (Icegraph 2.0 http://iceweb1.cis.ec.gc.ca/IceGraph/) are representative of the sea ice conditions experience by seals in and around Arviat where presumably a substantial number of their harvest samples came from (see later note on need for samples sizes from each study site)? Did the authors examine sea ice trends in NW Hudson Bay which would be more representative of the Arviat sample? How do these sea ice trends compare to eastern Hudson Bay?
Lines 106-109: Given the geographic separation of the two harvest sites in Arviat and Sanikiluaq can the authors (1) provide data on the numbers of seals harvested from each site in each year, and (2) a statistical comparison of the variables measured in the two sampling locations across years to ensure that the data indeed can be pooled.
Line 141-142: “In addition, body condition significantly decreased with increasing open water period (t = -2.0, p < 0.05)…” Given that open water period represents not only early spring breakup but also late fall freeze-up wouldn’t it be more biologically relevant to assess current body condition in relation to past environmental condition (e.g. just sea ice breakup) as opposed to both past and future conditions (i.e. the total duration of the ice free period in the year of harvest)?
Line 143-145: Did the authors test for lag effects in environmental conditions and their influence on demography and body condition in subsequent years?
Line 153-155: Similar to my comment above why not test for a lag in the relationship between cortisol levels and ovulation rates in seals the following year? It is easy to state “The highest stress levels occurred in 2010, and the lowest recorded ovulation rates occurred in 2011 which supports the pattern of a decrease in ovulation rate after the record high stress levels.” without showing the pattern….
Line 189: Can the authors clarify whether they are referring to decreases in the physiological and energetic demands of prey or ringed seals?
Line 463: Figure 3 plot A the X axis is labeled as mean of ice-free season duration. Isn’t this an exact value calculated subtracting the breakup and freezeup dates? How the duration of open water period is never actually defined in the manuscript…..

Additional comments

The following are general comments for the author:

Line 38: “….populations suffer demographic mortality.” Can mortality be non-demographic? I suggest removing the word demographic here as it is redundant.
Line 42: “…demographic mechanism of population decline…” should be mechanisms of population decline as concurrent declines in reproduction and survival could result in population declines.
Line 46-49 reads: “Few studies have linked marine mammal demographic responses to climate change (Poloczanaska et al. 2007) with the notable exception of ringed seals (Meier et al. 2004, Post et al. 2009), where the majority of research results reflect changes in foraging behaviour (Young and Ferguson 2014, Hamilton et al. 2015).” The subject of this sentence is demographic responses of marine mammals to climate change which may be related to changes in food availability but neither Young and Ferguson nor Hamilton et al. provide links between foraging behavior and demography. I suggest this sentence be reformatted. In addition the authors have failed to acknowledge a substantial body of literature linking polar bear demography and climate change (e.g. Regehr et al. 2007, Hunter et al. 2010 etc.).
Line 69-71: Can the authors provide literature to support these statements?
Line 162: Derocher et al. 2004 is a review paper. You should cite the original paper which in all fairness gives credit to the authors that did the work as was done for Chambellant et al. 2012.
Line 167: “However, no research results have suggested short-temporal pulses in condition and abundance for either seals or polar bears in the Hudson Bay ecosystem, although a regime shift likely occurred in late 1990s (Gaston et al. 2012).” Given that no abundance data are presented, other than percent pups in the harvest I suggest this sentence be reworded as there has been no demonstration of a significant reduction in sub-adult or adult survival which would ultimately drive any abundance trends observed.
Line 448: Where do the data come from for the “….possible years with poor ringed seal condition: 1974/74, 1983, 1992,1998,2010.”?

·

Basic reporting

Line 61-71: This paragraph needs further citations to back the statements. Everything written here appears to be factually correct, but the introduction should include the relevant literature to support the statements of what is known.

Line 74-76: The adjective “south*” is used three times in the same sentence. Does Southern Hudson Bay refer to a distinct population of ringed seals or a specific geographical area? It may be easier to state Hudson Bay rather than split hairs about the exact area, especially as the sea ice study area is later defined to be “eastern Hudson Bay” (Line 95).

Line 72-80: More information on sea ice loss/change would aid in providing context to readers not familiar with the system. There is an assumed relationship between environmental change and sea ice, but this is not stated explicitly. The single reference to Post et al 2013 is insufficient. Additionally, I think it is worthwhile to the reader to state that this region is on the edge of the maximum sea ice extent, and goes through a seasonal cycle of complete ice gain and loss.

Line 81-91: Reorganize and rewrite for clarity. This paragraph is difficult to comprehend on first pass and takes a couple reads. I suggest starting with the overall objective, simply stated as comparing indices in the productivity and health of the ringed seal population with environmental covariates over time. Then specify the dependents as you have, and finally make predictions based on overall trend and punctuated short term changes.

METhODS: A study area figure is needed, with the town sites where seal data was collected in reference to the areas in which sea ice data was inferred from.

Line 188: Remove the comma after “have”

Experimental design

Why was the sea ice data only from eastern Hudson Bay used (line 95), when the seal data was collected from both the east (Sanikiluaq, NU, Canada) and the west (Arviat, NU, Canada)? Some justification for this is needed in the methods.

While the authors have included some basic environmental metrics, as a reader, I was hoping for a more diverse set of independent variables to compare with such a rich and rare dataset on ringed seals. What about temperature? Rainfall? Snow accumulation? Is there productivity data for the Hudson Bay? Is there an ability to parse the data between the two harvest sites (east vs. west) and see if the response was homogenous across Hudson Bay, or if there was significant within ecoregion response?

Line 128: “NAO was highly correlated with AO (0.8), thus was removed from all analyses”. Two points: 1) The sentence as written implies the NAO was removed not the AO; and 2) How was it decided that AO should be removed while the NAO remain?

Line 122-131: Modelling, two points:

1) Because these are time-series analyses, the authors need to be wary of non-stationarity in their data, which can artificially inflate the significance of their findings. I am not arguing against using linear models, but if they do, they should check that autocorrelation of the residuals is not a factor. I suggest using a Durbin-Watson test (R: https://cran.r-project.org/web/packages/car/index.html). If serial correlation is an issue for any of their dependent variables, then more complex models may be needed, such as an autoregressive integrated moving average (ARIMA) model (R: https://stat.ethz.ch/R-manual/R-devel/library/stats/html/arima.html).

2) The authors used a general linear model, and did not account for the potential differences between sample sites. The data is structured between the two sites, and it cannot be assumed that the response is necessarily homogenous across a large ecosystem, that the harvest methods are identical, or that the samples are divided evenly between the sites. I recommend that a mixed model with a random intercept for sample location be used to see if it describes more of the variation in the dataset than a fixed-effect approach. A mixed model based on sample location may not improve the fit, but this should be checked using information criterion and reported.

Validity of the findings

Line 153 / Line 172: While I agree it is appropriate to state that 2010 cortisol levels had the highest level of variation, I think the authors have to be careful about stating that the cortisol / stress levels were the “highest”, “record high” or “increased” in 2010, which is done both in the Results and Discussion. Yes, it is obvious that some individuals had a very high stress response in 2010, but some individuals also did not, and there are no stats to back the claim that the mean response in 2010 was significantly different from any other year. As the paper is discussing the population response, I recommend rephrasing the cortisol results to just focus on the wide variation. That in and of itself tells a neat story.

DISCUSSION: It would be good to see a discussion on how episodic changes are linked or not linked with potential systemic changes. There is evidence from other well-studied systems, such as the Beaufort Sea, of cyclic/episodic changes in sea ice conditions causing punctuated changes in the productivity of ringed seal populations. It is unclear whether by episodic the authors meant recurring/cyclic or stochastic event. How we as ecologists define an event being part of systemic change or a one-off stochastic event is vital to how we interpret alterations due to climate change in the systems we study (noise vs. signal). There is the potential for a really nice discussion of episodic versus systemic change in a sea ice ecosystem within the context of climate change using this dataset as an example.

DISCUSSION: The mechanism of an earlier breakup and longer open water season causing punctuated declines in ringed seal productivity is a novel finding. In previous research of episodic crashes in the Beaufort Sea, heavy ice and delayed breakup were responsible for punctuated declines in ovulation and pup production (as noted on Lines 235-237). Can the authors potentially explain the mechanisms behind this discrepancy?

Line 162: There are newer assessments on polar bear productivity in Hudson Bay in relation to sea ice than Derocher et al 2004.

Examples:
Regehr E.V., N.J. Lunn, S.C. Amstrup, and I. Stirling. 2007. Effects of earlier sea ice breakup on survival and population size of polar bears in western Hudson Bay. J Wildl Manag 71:2673–2683.

Lunn N.J., S. Servanty, E.V. Regehr, S.J. Converse, E. Richardson, and I. Stirling. 2016. Demography of an apex predator at the edge of its range: impacts of changing sea ice on polar bears in Hudson Bay. Ecol Appl 26:1302–1320.

Obbard M.E., M.R.L. Cattet, E.J. Howe, K.R. Middel, E.J. Newton, G.B. Kolenosky, K.F. Abraham, and C.J. Greenwood. 2016. Trends in body condition in polar bears (Ursus maritimus) from the Southern Hudson Bay subpopulation in relation to changes in sea ice. Arctic Science 2: 15–32.

Line 173 – 185: A further comparison to other systems is needed here. Lois Hardwood has recently published a paper on the links between species dependent on polar cod and the trends in productivity and condition in the Beaufort Sea. This would be some nice supporting evidence for the mechanism that the authors are suggesting here.

Link to paper: http://www.sciencedirect.com/science/article/pii/S0079661115001007?np=y

Line 201 – 218: The implied mechanism of a disease is interesting, but the authors need to make this speculation more clearly. The paragraph ends with discussing whether what happened recently in the Beaufort Sea had a cause or not. This leaves the reader to make the jump between environmental change and a possible disease outbreak in Hudson Bay, without clear intention from the authors. If the authors want to go this route, I think it is fine to speculate, but further information is required on how this might occur, perhaps drawing on examples from other ecosystems. The current paragraph only makes a vague association between similar behaviors of seals in both systems.

Line 225-226 & Supplementary Material: The delayed/incomplete molt is an interesting observation. Because ringed seals haul out in great number in June to molt, could the lack of a resting platform (early sea ice breakup) potentially affected their molt (maybe incomplete) and perhaps their thermoregulation? Could hyperthermia be a potential mechanism for the observations of lethargy and use of tidal flats?

SUPPLEMENTARY TABLE 2: Fantastic inclusion of local reporting and knowledge. Although not enough on it’s own, when combined with the biological and environmental data, it tells a compelling story.

Additional comments

Continuous time series on the condition and productivity of any Arctic mammal is a rare and unique opportunity. This dataset is one that I am keen to see published, as well as the inference into the mechanisms of change. I applaud the authors on gathering a wide array of biological information as well as local knowledge to try to maximize their ability to detect population-level change in a remote and difficult setting to work in, which is also undergoing some of the most rapid change anywhere on the planet.

However, while the dataset is impressive, I also feel that it is deserving of a more robust analysis and discussion. The inclusion of only four independent variables in the models, including only one ice metric, stops short of a thorough investigation into the possible mechanisms of change. The Discussion is limited by an intensive focus on previous ringed seal work within Hudson Bay, and at times makes a rather pedestrian pass at inferring possible mechanisms by describing the results rather than discussing them in a contextual way. I welcome another read of this manuscript, but only after considerable attention has been given to investigating and discussing the mechanisms of change.

External reviews were received for this submission. These reviews were used by the Editor when they made their decision, and can be downloaded below.

---

## Round 0.2 · Minor Revisions

· Academic Editor

Minor Revisions

The reviewers were appreciative of the improvements you have made to the MS. Both have smaller concerns that I would like to see addressed in a final revision before publication.

Reviewer 1 ·

Basic reporting

see below

Experimental design

see below

Validity of the findings

see below

Additional comments

The authors have responded to most of my concerns. Only three housekeeping issues remain.
1. Although the authors indicate that they have added text to explain how body condition is determined there is no such description anywhere in the methods (or anywhere else) of the revised manuscript. Please add the description to the revised version - this is an rather important component of your analysis and it is still not explained.

2. For what are now lines 174-175 my original question still stands: Why do you use 2011 as the lower reference (~45%) here when 2012 is even lower at ~40%?
You indicate in your response that there was an error in the figure which explains this but Figure 2C is unchanged from the previous version and according to the newly added table 2012 is lower than 2011.....

3. Again I am confused by Figure 3B – why are there still only 9 data points? According to Table 1 there should be 10 points. It appears that the body condition from 2012 (the lowest value) is still missing. Your response to my original observation was that there was an error in the plot but the plot remains identical to the original in the revised version. Please fix or explain what happened to 2012 (and verify that it was included in your analyses).

·

Basic reporting

See Below

Experimental design

See Below

Validity of the findings

See Below

Additional comments

The authors have done a fantastic job of broadening the discussion and clarifying the methods and data sources. They have adequately answered all of my inquiries, and made the changes necessary to the manuscript. I would still appreciate a study area figure, but I leave that to the choice of the Editor and Authors.

One minor comment:
Line 175 says blubber mass increased to about 55% in 2013, but Table 1 shows precent fat as 48.1% in 2013. As you have the specific numbers in Table 1, I would update this sentence to reflect those values, rather than say "about".

External reviews were received for this submission. These reviews were used by the Editor when they made their decision, and can be downloaded below.

---

## Round 0.3 · accepted · Accept

· Academic Editor

Accept

Thank you for addressing the remaining concerns of the reviewers. I look forward to seeing your manuscript in print.

External reviews were received for this submission. These reviews were used by the Editor when they made their decision, and can be downloaded below.